# Model-based targeted dimensionality reduction for neuronal population data

**Mikio C. Aoi**[*]
Princeton Neuroscience Institute
Princeton University,
Princeton, NJ 08544
maoi@princeton.edu

**Jonathan W. Pillow** [†]
Princeton Neuroscience Institute
Princeton University,
Princeton, NJ 08544
pillow@princeton.edu

## Abstract

Summarizing high-dimensional data using a small number of parameters is a ubiquitous first step in the analysis of neuronal population activity. Recently developed methods use "targeted" approaches that work by identifying multiple, distinct low-dimensional subspaces of activity that capture the population response to individual experimental task variables, such as the value of a presented stimulus or the behavior of the animal. These methods have gained attention because they decompose total neural activity into what are ostensibly different parts of a neuronal computation. However, existing targeted methods have been developed outside of the confines of probabilistic modeling, making some aspects of the procedures ad hoc, or limited in flexibility or interpretability. Here we propose a new model-based method for targeted dimensionality reduction based on a probabilistic generative model of the population response data. The low-dimensional structure of our model is expressed as a low-rank factorization of a linear regression model. We perform efficient inference using a combination of expectation maximization and direct maximization of the marginal likelihood. We also develop an efficient method for estimating the dimensionality of each subspace. We show that our approach outperforms alternative methods in both mean squared error of the parameter estimates, and in identifying the correct dimensionality of encoding using simulated data. We also show that our method provides more accurate inference of low-dimensional subspaces of activity than a competing algorithm, demixed PCA.

## 1 Introduction

Neuroscience has recently seen a massive expansion in the number of neurons that can be recorded from a single animal, largely due to transformative technological advancements in electrode design and two-photon imaging. One of the effects of our increased measurement capacity is an increased interest in the properties of the activity of groups of neurons (i.e. *population* activity), as opposed to analyzing the activity of single-neurons independently [1]. One goal of analyzing population activity is to characterize the ways in which groups of neurons coordinate to perform task-relevant computations.

Dimensionality reduction is central to the analysis of population activity [1]. Concomitant with the broader use of classical dimensionality reduction methods like PCA and ICA come the recognition that these methods often do not take full advantage of well-characterized properties of neuronal population data such as tensor structure or temporal correlations in spike rates and a number of recent data analysis techniques have been developed to improve our ability to meet such specific challenges

---

[*]http://mikioaoi.com
[†]http://pillowlab.princeton.edu/jpillow/

[2–7]. Of particular interest have been methods of dimensionality reduction for population data that distinguish between the effects of various inputs and outputs, or "task variables," such as a stimulus strength, an experimental context, or a behavioral outcome [8–11]. We will refer to these methods collectively as "targeted" methods.

Although several targeted methods of dimensionality reduction exist, two recent methods stand out among existing methods: demixed principle components analysis (dPCA) [8] and targeted dimensionality reduction (TDR) [9]. Both of these methods were developed for the analysis of neuronal population data that inherently have observations of neuronal activity structured as matrices (ex. neurons by row, time by columns) and both attempt to identify low-dimensional subspaces that best describe the population responses to an individual task variable.

The most recent version of dPCA [8] is a general method with relatively weak modeling assumptions, arbitrary dimensionality, and a fast estimation algorithm based on low-rank regression [12]. However, dPCA requires that all observed neurons display firing rates for all possible combinations of task variables, a condition that may be too strict to be applicable for complex experiments. In contrast, TDR [9] utilizes a linear regression-based approach that circumvents the need to have observed every neuron at every combination of task variables by imposing an explicit relationship between regressors and outputs. However, the TDR method is limited to a one-dimensional subspace per task variable. It is not clear that only one dimension is sufficient to describe the population activity associated with a given task variable. For example, sequential activation of neurons during decision making has been observed in rodents, where the precise ordering of activations depends on which decision the animal makes [13] and population code "morphing" has been observed in monkeys where decision encoding changes over time [14]. These types of dynamic encoding schemes are inherently high-dimensional and any method constrained to too-few dimensions will fail to fully characterize such activity. Lastly, none of the existing methods have principled approaches to identifying the dimensionality of the data, making post hoc analysis particularly difficult.

Here we propose a model-based method for targeted dimensionality reduction based on an extension of the framework proposed by [9]. Our approach overcomes a number of the drawbacks of existing methods. Using a probabilistic generative model of the data, we can infer the optimal dimensionality of the low-dimensional subspaces required to faithfully describe underlying latent structure in the data. In the following, we describe the model, which we call model-based targeted dimensionality reduction (MBTDR), its assumptions, and an efficient estimation procedure for model parameters and dimensionality. We then demonstrate the accuracy of our estimation algorithm against alternative methods of estimation.

## 2 Explicitly low-dimensional model of population activity

### 2.1 High-dimensional description

Our model begins with a description of trial-by-trial neuronal activity in terms of a linear regression with respect to the task variables. We assume that the activity $y_{i,k}(t)$ of the $i^{\text{th}}$ neuron at time $t$ on trial $k$ can be described by a linear combination of $P$ task variables $x_k^{(p)}$, $p = 1, \ldots, P$ (ex. stimulus variables, behavioral outcomes, and nonlinear combinations), such that

$$y_{i,k}(t) = x_k^{(1)}\beta_{i,1}(t) + x_k^{(2)}\beta_{i,2}(t) + \cdots + x_k^{(P)}\beta_{i,P}(t) + \epsilon_{i,k}(t).$$

where the values of the $P$ task variables $x_k^{(p)}$ are known, the $\beta_{i,p}(t)$ are unknown coefficients, and $\epsilon_{i,k}(t)$ is noise. This basic model structure is identical to that of the regression model used in [9] and has been successfully employed in characterizing neuronal activity of single neurons [15].

To represent all neurons simultaneously, we simply concatenate all $i = 1, \ldots, n$ responses into a vector and write

$$\mathbf{y}_k(t) = x_k^{(1)}\boldsymbol{\beta}_1(t) + x_k^{(2)}\boldsymbol{\beta}_2(t) + \cdots + x_k^{(P)}\boldsymbol{\beta}_P(t) + \boldsymbol{\epsilon}_k(t),$$

where $\mathbf{y}_k(t) = (y_{1,k}(t), \ldots, y_{n,k}(t))^\top$, $\boldsymbol{\beta}_p(t) = (\beta_{1,p}(t), \ldots, \beta_{n,p}(t))^\top$, and $\boldsymbol{\epsilon}_k(t) = (\epsilon_{1,k}(t), \ldots, \epsilon_{n,k}(t))^\top$.

Neuronal recordings are often performed in experiments where trial epochs are of fixed duration. We can take advantage of this structure by regarding the observation on each trial to be

a matrix, $\mathbf{Y}_k = (\mathbf{y}_k(1), \ldots, \mathbf{y}_k(T))$, which is a linear combination of $P$ coefficient matrices $\mathbf{B}_p = (\boldsymbol{\beta}_p(1), \ldots, \boldsymbol{\beta}_p(T))$ giving the observation model

$$\mathbf{Y}_k = x_k^{(1)}\mathbf{B}_1 + x_k^{(2)}\mathbf{B}_2 + \cdots + x_k^{(P)}\mathbf{B}_P + \mathbf{E}_k, \tag{1}$$

where $\mathbf{E}_k = (\boldsymbol{\epsilon}_k(1), \ldots, \boldsymbol{\epsilon}_k(T))$. A schematic illustration of this basic setting is shown in Figure 1.

In general, not all neurons are observed simultaneously. Most often they are observed sequentially or in sequential blocks. Suppose we do not observe all rows of $\mathbf{Y}_k$ on all trials but instead observe $n_k \leq n$ neurons. If we let $\mathbf{Y}_k$ be a latent matrix of all recorded neurons from all trials, then we can describe the observed neurons on any given trial by $\mathbf{Z}_k = \mathbf{H}_k\mathbf{Y}_k$, where $\mathbf{H}_k$ is a $n_k \times n$ matrix where each row is a 1-hot vector providing the index of an observed neuron.

## 2.2  Low-dimensional description of observations

With no additional constraints our observation model (1) is extremely high dimensional and is effectively a separate linear regression for each neuron at every time point. This would only be a sensible model if we believed that neurons were not in fact coordinating activity between each other or across time. We would like to be able to express the prior belief that there are correlations across the population but that correlations in activity due to different values of stimuli are not necessarily the same as those due to the behavior of the animal.

To accomplish this we can describe each characteristic response matrix $\mathbf{B}_p$ by a low-rank factorization, i.e. $\mathbf{B}_p = \mathbf{W}_p\mathbf{S}_p$, where $\mathbf{W}_p$ and $\mathbf{S}_p$ are $n \times r_p$ and $r_p \times T$ respectively, where $r_p = \text{rank}(\mathbf{B}_p)$. Equivalently, we can say that $r_p$ is the dimensionality of the encoding of task variable $p$. This formulation has an intuitive interpretation, illustrated schematically in Figure 1A: the characteristic response $\beta_i^p(t)$ of each neuron to the $p^{\text{th}}$ task variable can be expressed as a linear combination of $r_p$ weighted basis functions $\beta_i^p(t) = \sum_{j=1}^{r_p} w_{i,j}^{(p)} s_j^{(p)}(t)$, where $r_p$ is the dimensionality of the encoding, $\{s_j^{(p)}(t)\}_{j=1}^{r_p}$ are a common set of time-varying basis functions, and $\{w_{i,j}^{(p)}\}_{j=1}^{r_p}$ are neuron-dependent mixing weights.

The example in Figure 1A displays a model with two task variables $(x_1, x_2)$, where the $x_1$ subspace is 1D and the $x_2$ subspace is 2D. The columns of $\mathbf{W}_p$'s weight each time-varying basis function differently for each neuron. Collectively, these weights define the subspace of activity that encodes task variable $x_p$. For $x_1$, the encoding is 1D because only one basis function is needed to describe the population response to $x_1$. The $x_2$ response is slightly more complex, with different responses at different times, requiring at least two basis functions.

# 3  Model estimation

The goal of inference is to estimate the factors of $\mathbf{B}_p$ and the ranks $r_p$. Our proposed estimation strategy is to estimate one set of factors ($\{\mathbf{W}_p\}$ or $\{\mathbf{S}_p\}$) while integrating out the other. For example, if we define a prior over the mixing weights $\{\mathbf{W}_p\}$ denoted by $p(\mathbf{W})$, and a data likelihood $p(\mathbf{Z}|\mathbf{W}, \mathbf{S})$ then the marginal likelihood of the matrix of time-varying basis functions $\mathbf{S}$ can be obtained by

$$p(\mathbf{Z}|\mathbf{S}, \boldsymbol{\lambda}) = \int_{-\infty}^{\infty} p(\mathbf{Z}|\mathbf{W}, \mathbf{S}, \boldsymbol{\lambda})p(\mathbf{W})d\mathbf{W}.$$

In principle, either set of factors may be selected. In practice however the set of factors with lowest dimension should be selected to keep computational costs low. In this paper we focus on the case where $T \ll n$ and we therefore will estimate $\{\mathbf{S}_p\}$ while integrating over $\{\mathbf{W}_p\}$. The fact that either set of factors may be determined in this way means that there is a duality between rows and columns imposed by this model that is similar in principle to the duality between factors and latent states for probabilistic principle components analysis [16].

In practice inference can be considerably simplified if we let the noise distribution and prior distribution of $\mathbf{W}$ both be Gaussian, which permits closed-form expression of the marginal and posterior densities. We will let all elements of $\mathbf{W}$ to be independent standard normal, (i.e., $\mathbf{w} \sim \mathcal{N}(0, I_{\tilde{r}n})$ where $\tilde{r} = \sum_p \text{rank}(\mathbf{B}_p)$). In addition, we let the noise covariance on all trials be given by $\mathbf{E}_k \sim \mathcal{MN}(0, \mathbf{D}^{-1}, I_T)$, where $\mathcal{MN}(M, A, B)$ denotes the matrix normal distribution with row covariance $A$ and column covariance $B$, and $\mathbf{D} \equiv \text{diag}(\lambda_1, \ldots, \lambda_n)$ where $\lambda_i$ is the inverse noise

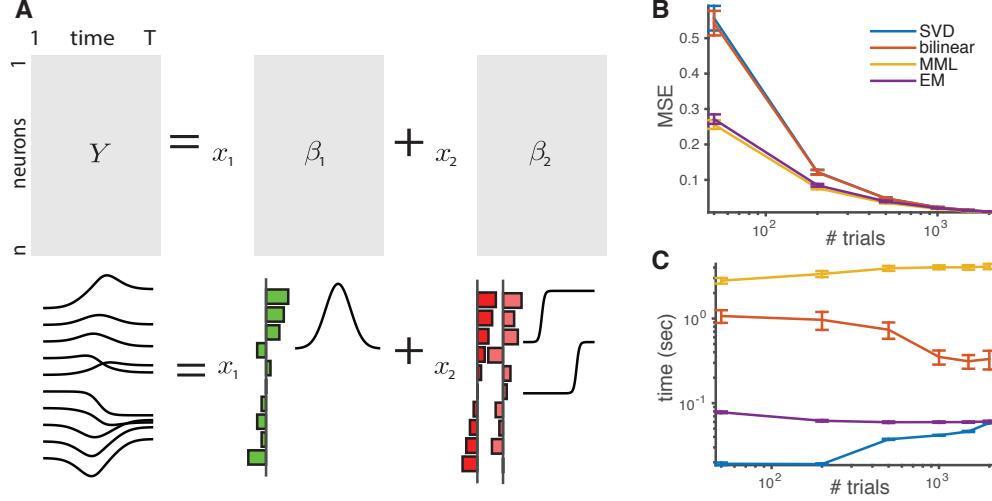

**Figure 1: A**: Schematic illustration of low-rank regression model. The $n \times T$ response matrix can be decomposed into two response matrices $(\boldsymbol{\beta}_1, \boldsymbol{\beta}_2)$, each corresponding to one task variable (upper panel). Each response matrix can be factorized into a small number of row and columns vectors, making the population response a linear combination of a small number of common basis functions weighted differently for each neuron. **B**: Results of simulation study evaluating parameter estimation accuracy for different estimation procedures. Legend indicates the method used. Abscissa indicates the number of trials used for the simulations. Error bars indication 95% confidence intervals over 100 runs. **C**: Duration of computation for methods and trials used in **B**.

variance of neuron $i$. We therefore assume that the weights are a priori independent and that the noise variance is independent across both neurons and time. In principle, our framework supports the application of more structured priors and noise covariances but we will save the exploration of more elaborate models for future work.

### 3.1 Marginal likelihood of timecourses S

Since our model is linear and Gaussian, the marginalized density $p(\mathbf{Z}|\mathbf{S}, \boldsymbol{\lambda})$ is also Gaussian and can be easily derived using standard Gaussian identities [17]. However, a naive derivation of the marginal likelihood requires the log determinant and inverse of a matrix which is $\tilde{N}T \times \tilde{N}T$, where $\tilde{N} = \sum_i N_i$, such that $N_i$ is the number of observed trials for neuron $i$. Thus, if all neurons are observed on all trials, then the dimensions of the marginal covariance will be $nNT \times nNT$, which can be prohibitively large for even moderately sized datasets since the determinant and inverse in general will have computational complexity $\mathcal{O}(\tilde{N}^3 T^3)$. Luckily, the expression for the marginal likelihood can be dramatically reduced by exploiting the factorization of regression parameters.

If we let $\mathbf{S} \equiv \mathrm{blkdiag}(\mathbf{S}_1, \ldots, \mathbf{S}_P)$, and $\boldsymbol{\lambda} = (\lambda_1, \ldots, \lambda_n)$ then we can derive (see Supplementary Material for details) the following expression for the marginal likelihood in terms of $\mathbf{S}$ and $\boldsymbol{\lambda}$,

$$\ell(\mathbf{s}, \boldsymbol{\lambda}) = -\frac{1}{2} \left( \tilde{n}T \log 2\pi + \sum_{i=1}^{n} \left( -N_i T \log \lambda_i + \lambda_i \mathbf{y}_i^\top \mathbf{y}_i + \log |\mathbf{C}_i| - \lambda_i^2 \mathrm{Trace}[\mathbf{R}_i \mathbf{S}^\top \mathbf{C}_i^{-1} \mathbf{S}]) \right) \right). \tag{2}$$

where the matrices $\mathbf{R}_i$ and $\mathbf{C}_i$ are defined by

$$\mathbf{R}_i = (\mathbf{X}_i^\top \otimes I_T) \mathbf{y}_i \mathbf{y}_i^\top (\mathbf{X}_i \otimes I_T), \qquad \mathbf{C}_i = \lambda_i \mathbf{S}(\mathbf{A}_i \otimes I_T) \mathbf{S}^\top + I_{\tilde{r}}, \tag{3}$$

respectively, where $\mathbf{X}_i$ is the $N_i \times P$ design matrix that only includes trials where neuron $i$ was observed, $\mathbf{A}_i = \mathbf{X}_i^\top \mathbf{X}_i$ and $\mathbf{y}_i = \left(\mathbf{y}_i^{1\top}, \ldots, \mathbf{y}_i^{N\top}\right)^\top$, with $\mathbf{y}_i^k$ being the length-$T$ response of neuron $i$ on trial $k$.

The expression in (2) reveals two things about the structure of dependencies within the model. First, we notice that the likelihood factorizes over neurons, making evaluation of the likelihood potentially

highly parallelizable. Second, the trace term is remenicent of the quadratic term of a matrix normal model, indicating that we can intuitively think of the posterior covariance $\mathbf{C}_i$ and the rank-1 matrix $\mathbf{R}_i$ as the neuron-dependent contributions to the row and column covariances of $\mathbf{S}$, respectively.

Maximum marginal likelihood (MML) estimates for $\mathbf{S}$ and $\lambda_i$ can be obtained by directly maximizing (2) by gradient ascent.

## 3.2 Posterior distribution of neural weights W

Once an estimate of $\mathbf{S}$ and $\boldsymbol{\lambda}$ is obtained we can do posterior inference on $\mathbf{W}$. Because our model is linear and Gaussian, the posterior density $p(\mathbf{W}|\mathbf{Z}, \mathbf{S}, \boldsymbol{\lambda})$ is also Gaussian and admits closed-form expressions for the posterior expectation and variance of $\mathbf{W}$. Because of our low-rank model structure, the posterior of the weight matrices $\{\mathbf{W}_p\}$ factorizes over neurons and we can estimate the weights $\mathbf{W}$ for each neuron separately and achieve computational savings relative to joint estimation over all neurons simultaneously.

We can define a $\tilde{r} \times 1$ vector $\boldsymbol{\omega}_i$ that contains all of the weights for neuron $i$. Collectively, the $\boldsymbol{\omega}_i$ can be expressed as

$$
\begin{pmatrix} \boldsymbol{\omega}_1 \\ \vdots \\ \boldsymbol{\omega}_n \end{pmatrix} = \mathrm{vec} \begin{pmatrix} \mathbf{W}_1^\top \\ \vdots \\ \mathbf{W}_P^\top \end{pmatrix}.
$$

This notation allows us to do efficient posterior inference over $\boldsymbol{\omega}_i$, where the posterior expectation and covariance of $\boldsymbol{\omega}_i$ are given by

$$
\mathbb{E}_{\boldsymbol{\omega}_i|\mathbf{S},\mathbf{Z}}[\boldsymbol{\omega}_i] = \lambda_i \mathbf{C}_i^{-1} \mathbf{S} (\mathbf{X}_i^\top \otimes I_T) \boldsymbol{\zeta}_i, \qquad \mathrm{Cov}_{\boldsymbol{\omega}_i|\mathbf{S},\mathbf{Z}}[\boldsymbol{\omega}_i] = \mathbf{C}_i^{-1},
$$

where $\mathbf{C}_i$ is defined as in (3).

## 3.3 Decoding

Once estimates of $\mathbf{B}_p$ are obtained we can decode new trials using the observation likelihood. This is a distinct feature of our method that is not available to dPCA and TDR. The former methods are used for estimation of the encoding but must learn a separate decoder to decode task variables from the activity. Because of the probabilistic formulation of our model we can do encoding and decoding within the same framework, allowing us to directly ask questions about how the structure of the encoding influences the ability of down-stream populations to decode the information in the recorded population. While we do not pursue decoding further in this paper we included a description of the optimal decoder in the Supplementary Material.

# 4 ECME algorithm for parameter estimation

In general, maximization of the marginal likelihood (2) can be relatively slow when the number of parameters is large. We therefore derive a "expectation-conditional maximization, either" algorithm (ECME) [18] where parameters are block-wise estimated by either maximizing the conditional expectation of the complete data log likelihood or the marginal likelihood. Our algorithm has closed-form updates for each parameter block.

Note that, for Bayesian linear regression with Gaussian likelihood and prior, an otherwise unstructured model would have, for $M$ parameters, an ECME update with computational complexity of $\mathcal{O}(M^3)$. In contrast, due to the additional low-rank structure of our model, and despite each M-step updating $\tilde{r}T + n$ parameters, our M-step updates have computational complexity $\mathcal{O}(\tilde{r}^3)$, where there are typically $\tilde{r} \ll \min\{n, T\}$. This means that the actual computational cost of ECME is limited only by the underlying dimensionality of the data, and not to the total number of parameters per se.

As we demonstrate in Section 6.1, while our EM algorithm provides parameter estimates that are only slightly worse in mean-squared error as maximizing the marginal likelihood directly, this small additional error has a serious impact on dimensionality estimation. We therefore use our ECME algorithm to provide fast, high-quality initialization for maximizing the marginal likelihood by gradient descent.

# 5 A greedy algorithm for rank estimation

While our model can identify subspaces of any dimension up to $D_{\max} = \min\{n, T\}$, the dimensionality of each subspace must be specified *a priori*. Although we may use standard model selection techniques to compare the goodness of fit between models with alternative configurations an exhaustive search over all possible models would require searching over $D_{\max}^P$ possible configurations. We therefore developed a greedy algorithm for estimating the optimal dimensionality. A summary of the procedure is presented in Algorithm 1.

Recall that the dimensionality of each task-variable encoding corresponds to the rank of each $\mathbf{B}_p$. We begin the algorithm by first estimating the model parameters with rank $r_p = 1$ for all $p$ (although in principle we may start at $r_p = 0$, denoting the null model for all elements of $\mathbf{B}_p$), giving us a model with total dimensionality $\tilde{r} = \sum_{p=1}^{P} r_p$ and at the first iteration $\tilde{r}_1 = P$. At the $j^{\text{th}}$ iteration, we estimate the parameters of $P$ models, where each model has the dimension of one of the task variables increased by 1, while keeping all other dimensionalities the same as in the previous iteration. We then have $P$ models, each with total dimensionality $\tilde{r}_{j+1} = \tilde{r}_j + 1$. We then evaluate the AIC of each of these $P$ models and keep the model that displayed the greatest decrease in AIC relative the the previous iteration for the next iteration. In this way we grow the total dimensionality of the model by one on each iteration. The algorithm is formally outlined in Algorithm 1. [3]

---
**Algorithm 1** Estimation of dimensionality
---
Let $\mathbf{r} \equiv (r_1, \ldots, r_P)$, $\mathbf{e}_p$ is the elemental vector, AIC($\mathbf{r}$) is the Akaike information criterion for a model with ranks $\mathbf{r}$

1: **procedure** DIMEST($\mathbf{r}_0$,Data)
2:     $\mathbf{r} \leftarrow \mathbf{r}_0$, $\text{AIC}_0 \leftarrow \text{AIC}(\mathbf{r}_0)$         ▷ Initialize
3:     **repeat**
4:         **for** $p = 1, \ldots, P$ **do**     ▷ Calculate AIC for +1 rank for each task variable
5:             $\text{AIC}_p \leftarrow AIC(\mathbf{r} + \mathbf{e}_p)$
6:         **end for**
7:         **if** There is no $p$ s.t. $\text{AIC}_p < \text{AIC}_0$ **then** Break
8:         **end if**
9:         $p^* \leftarrow \arg\min_p \text{AIC}_p$, $r_{p^*} \leftarrow r_{p^*} + 1$     ▷ +1 rank for variable that most decreases AIC
10:         $\mathbf{r}_0 \leftarrow \mathbf{r}$, $\text{AIC}_0 \leftarrow \text{AIC}_{p^*}$
11:     **until** There is no $p$ s.t. $\text{AIC}_p < \text{AIC}_0$     ▷ Stop when AIC can no longer be decreased
12:     **return** $r$
13: **end procedure**
---

# 6 Simulation studies

## 6.1 Evaluation of parameter estimation with simulated data

We applied our greedy algorithm on simulated data in order to determine if it could accurately recover the true ranks using $n = 100$ neurons and $T = 15$ time points. For each run of our simulations we first selected a random dimensionality between 1-6 for each of $P = 3$ task variables (two graded variables with values drawn from $\{-2, -1, 0, 1, 2\}$ and one binary task variable with values $\{-1, 1\}$). Using these dimensionalities, the elements of $\mathbf{W}_p$ and $\mathbf{S}_p$ were drawn independently from a $\mathcal{N}(0, 1)$ distribution. To give us heterogeneous noise variances, the noise variance for each neuron was drawn from an exponential distribution with mean parameter $\sigma^2 = 50$. The resulting average SNR for any one task variable was -0.26 ($\pm 0.75$, $\log_{10}$ units). We then simulated observations according to our model with varying numbers of trials ($N \in \{50, 200, 500, 1000, 1500, 2000\}$). In order to simulate incomplete observations, we set the probability of observing any given neuron on any given trial to $\pi_{\text{obs}} = .4$. While we conducted experiments with varying numbers of trials and observation probabilities we generally found that decreased observation probabilities acted effectively as a decrease in sample size with a concomitant decrease in estimation accuracy. The results were

not particularly sensitive to the precise observation probability in this regime and we report only the results for the settings listed above.

For each set of observations, we estimated the parameters of the model in one of three ways, which we describe below.

We consider the following four methods for parameter optimization:

*1. Linear regression and SVD.* The elements of $\mathbf{B}_p$ for all $p$ were estimated by linear regression for each neuron and time point independently. Each estimate of the complete matrix $\mathbf{B}_p$ could then be expressed by its singular value decomposition (SVD) as $\mathbf{B}_p = \mathbf{U}_p \mathbf{D}_p \mathbf{V}_p^\top$, where $\mathbf{D}_p$ is the $n \times T$ diagonal matrix of $d = \min\{n, T\}$ singular values. We then set the smallest $d - r_p$ singular values to zero with the resulting matrix of $r_p$ nonzero singular values denoted by $\mathbf{D}_p^{(r_p)}$. The rank-$r_p$ estimates of $\mathbf{W}_p$ and $\mathbf{S}_p$ are then given by $\mathbf{W}_p^{(r_p)} = \mathbf{U}_p \mathbf{D}_p^{(r_p)1/2}$ and $\mathbf{S}_p^{(r_p)} = \mathbf{D}_p^{(r_p)1/2} \mathbf{V}_p^\top$, with the corresponding rank-$r_p$ estimate of $\mathbf{B}_p$ given by $\mathbf{B}_p^{(r_p)} = \mathbf{W}_p^{(r_p)} \mathbf{S}_p^{(r_p)}$.

The corresponding likelihood is given by

$$\ell(\mathbf{B}_p | \mathbf{Z}, \mathbf{H}', \hat{\mathbf{D}}) \propto \sum_k \text{Trace}[(\mathbf{Z}_k - \sum_p x^{(p)} \mathbf{B}_p)^\top \mathbf{H}' \mathbf{D} \mathbf{H}'^\top (\mathbf{Z}_k - \sum_p x^{(p)} \mathbf{B}_p] \tag{4}$$

*2. Bilinear optimization.* After initializing with the rank-$r_p$ estimates of $\mathbf{W}_p$ and $\mathbf{S}_p$ from the SVD method, the parameters can be further refined by bilinear regression. On each iteration, the values of $\mathbf{W}_p$'s are fixed, which leads to closed-form updates for conditional maximum likelihood estimates of $\mathbf{S}_p$'s and vice versa. Thus, the algorithm will alternate between estimating $\mathbf{W}_p$s and $\mathbf{S}_p$s until convergence. The bilinear regression method uses the same likelihood as shown in (4).

*3. ECME.* As described in the Supplementary Material.

*4. Maximum marginal likelihood (MML).* After initializing with the ECME estimates of $\mathbf{W}_p$ and $\mathbf{S}_p$, we estimate $\mathbf{S}_p$ by maximizing the marginal likelihood given by (2). No estimation of the $\mathbf{W}_p$ factors is required since the marginal likelihood only depends on $\mathbf{S}_p$.

For each setting of trial number $N$, we repeated this process 100 times and evaluated how well our algorithm estimated the true model parameters. The results are presented in Figure 1B,C.

We found that ECME and MML both produced mean-squared error (MSE) that was substantially smaller at all sample sizes that either the SVD or bilinear methods. While the differences in MSE between the ECME algorithm and MML were small, Figure 1C shows that the ECME algorithm was substantially faster than either MML or bilinear regression.

## 6.2 Evaluation of dimension estimation with simulated data

For each of the 100 runs of our simulation experiments we also evaluated how well our algorithm estimated the dimension of the characteristic responses by evaluating the difference between the true and estimated dimension of each task variable and counting the number of times that difference was observed. The results are presented in Figure 2A.

We found that all four methods tended to under-estimate the dimensionality as the number of trials decreased but that this underestimation was less pronounced for the ECME and MML methods, for which the vast majority of estimates resulted in the correct ranks even in the case of $N = 50$. Note that not only is this half the number of trials as neurons but since each neuron was only observed on about 40% of the trials this gives an average of only 20 trials per neuron. Therefore, our procedure recovers the true rank of the model the vast majority of the time even under conditions of vary small trial number relative to the size of observations.

We were surprised that despite the modest difference in MSE between the ECME and MML estimation algorithms, the dimension estimation seemed to be sensitive to these differences, with the ECME performing worse than MML despite the fact that these methods are in theory maximizing the same objective function. Nevertheless we propose that, due to the ECME algorithm's superior speed, ECME be used as an efficient initializer for MML estimation. We found that initializing the rank estimates this way limits the use of MML for rank estimation to just a few iterations.

For neuroscience applications, the observed spike counts are better described by a Poisson distribution than by a Gaussian. We therefore evaluated the robustness of our algorithm to this type of model misspecification by performing the same dimensionality estimation experiment with 2000 trials with observations drawn from Poisson($\mathbf{y}(t)$) distribution at each time bin. Our results are virtually indistinguishable from experiments using Gaussian observations (Fig. 2B, dashed line).

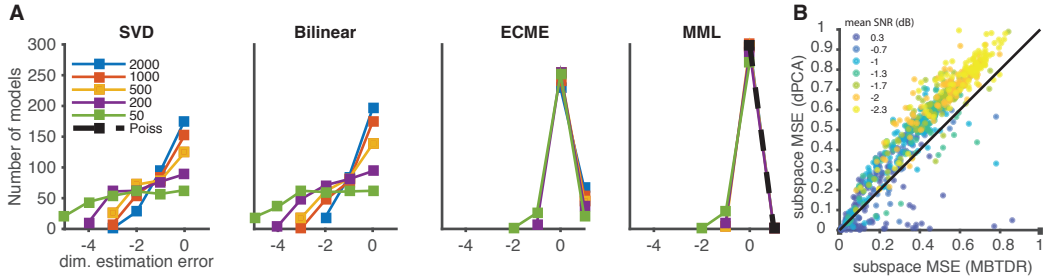

**Figure 2: Simulation studies**. **A**: Results of simulation study evaluating performance of Algorithm 1 for dimensionality estimation by means of different parameter estimation procedures. The legend indicates the sample size. Abscissa indicates the error in dimensionality estimate. Ordinate gives the number of estimated subspaces that obtained the corresponding error. Dashed line indicates model-mismatch experiment with Poisson observations and sample size 2000. **B**: Results of subspace estimation by our MML method compared with dPCA. The MML method out-performs dPCA at all but the highest SNR, where performance is similar.

# 7 Comparison with dPCA

## 7.1 Simulation experiments

The central goal of both our method and dPCA is to recover a basis that defines a set of low-dimensional subspaces that describe how the population varies with respect to each task variable (or pre-defined combination of task variables). In order to compare the quality of the subspaces identified by each method, we conducted a simple simulation study. The simulation setting was identical to those described in Section 6.1 using 100 trials per run except that, to keep the simulations as simple as possible, we defined just two binary task variables that were drawn randomly on each trial. The experiment was repeated for 100 runs.

We performed both dPCA and estimated the parameters on each run using MML and then compared the % mean-squared error between the true subspace and the estimated subspaces. We defined the true subspace based on the left singular vectors of the $\mathbf{B}_p$ matrices used for simulating the data. If $\mathbf{U}$ is the true subspace and $\hat{\mathbf{U}}$ is the estimated subspace then the % mean squared error is given by $\|\mathbf{U} - \hat{\mathbf{U}}\hat{\mathbf{U}}^\top\mathbf{U}\|_2^2/\|\mathbf{U}\|_2^2$.

The basis for the subspace estimated by MBTDR can be obtained by first estimating each $\mathbf{B}_p = \mathbf{WS}$, where $\mathbf{S}$ was estimated by MML and $\mathbf{W}$ was estimated by its posterior mean. We then used the left singular vectors of the estimated $\mathbf{B}_p$ to define the estimated basis. For the dPCA estimate, the analogous subspace is defined by their "encoding" subspace [8]. For both methods we assumed the correct dimensionality. We used the version of dPCA that is for non-sequential estimation and uses cross-validated regularization parameters. The results are presented in Figure 2B.

When the subspace is recoverable (i.e. principle angle is significantly less than 90 degrees), our method is virtually always closer to the true subspace. It is notable that the principle angle is an extremely sensitive measure of errors between subspaces and that both methods provide reasonable results when checked by eye. It is also notable that any differences are observable at all, which give us confidence that these results are quite strong.

# 8    Concluding remarks

We have introduced a new, model-based method to identify low-dimensional subspaces of neuronal activity that describe the response of neuronal populations to variations in task variables. We have also introduced a procedure of estimating both the parameters of this model and the dimensionality of each of the corresponding subspaces of activity. We compared our method in simulations to dPCA and showed that our method better recovers the low-dimensional subspace of activity for noisy data.

There are a number of additional advantages to using a model-based method for dimension reduction. The first is that our modeling framework is general enough that we could include even more structure to the model such as structured prior covariances and noise covariance. Our modeling approach also allows us to answer otherwise elusive questions about what quantities of the data are important. For example, virtually all other targeted methods effectively use peri-stimulus time histograms (PSTH's) as the sufficient statistics for subspace estimation. One interesting revelation of our model is that the PSTHs are *not* sufficient statistics. The sufficient statistics of our model are $(\mathbf{R}_i, \mathbf{A}_i, \mathbf{y}_i^\top \mathbf{y}_i, N_i)$ and these sufficient statistics can not be derived directly from the PSTHs. This suggests that methods that rely solely on the PSTHs may not be capturing important characteristics of the data.

## Acknowledgments

This work was supported by grants from the Simons Foundation (SCGB AWD1004351 and AWD543027), the NIH (R01EY017366, R01NS104899) and a U19 NIH-NINDS BRAIN Initiative Award (NS104648-01).

## Footnotes

[3]Demonstration code is available for download at the first author's website at http://www.mikioaoi.com/samplecode/RDRdemo.zip

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
