[Reviews · NeurIPS 2018]

Reviewer 1



Supervised dimensionality reduction has become a topic of interest in the systems neuroscience community over the last few years. Here, the authors suggested a very sensible extension to demixed PCA and targeted dimensionality reduction (TDR), which are recently developed but well-known and impactful methods in the field. From a modeling standpoint, I generally like the ideas behind this paper. However, I am disappointed that it heavily relies on simulated data rather than real biological datasets for its results. In particular, all datasets examined by the demixed PCA paper (in eLife) are freely available, so I feel that at least one of those datasets should have been analyzed for the purpose of comparison. I am not convinced that the proposed model would produce qualitatively different results from those already published. That being said, I think the proposed modeling framework is more straightforward than demixed PCA and offers the possibility of interesting future extensions. For example, one could incorporate a Poisson loss function or nonnegativity constraints into the model and deal with missing data in a straightforward manner. Thus, I think this deserves to be published as a worthy advance in the field, although I feel it is only marginally above the acceptance threshold due to limited empirical support on real data. Further development in this area would greatly improve the impact of the paper, especially if this method led to new scientific insights beyond what can be found with demixed PCA. The paper is of good quality and is clearly written. The ideas are original within the field of systems neuroscience; I wouldn't be surprised if other applied fields have developed similar methods (but I don't know of any). The ideas in this paper have the potential to be impactful if greater empirical support is shown on real data and if the authors provide well-documented and easy-to-use code to the community. Other comments: - The greedy method for estimating the dimensionality is sensible, and I can see why it could work well on simulated data. But I'm skeptical that it would unambiguously identify the "true" number of dimensions in real data where the model is misspecified. Choosing the number of components is always very challenging in real data and you can't trust everyone to choose this hyperparameter by the same procedure. Thus, an empirical demonstration that conceptual scientific insights are not overly sensitive to this hyperparameter choice would be a good addition to the paper. === New Comments After Author Feedback === I still feel that demonstrations on experimental data are critical to thoroughly evaluate this type of work. Thus, I am sticking to my score of "slightly above acceptance threshold". I think publishing this work will lead to interesting new discussions and models down the road. I congratulate the authors on their hard work thusfar and am looking forward to new developments.

Reviewer 2



The authors present an interesting and solid extension of the TDR method used in a number of neuroscience papers by introducing an assumption-free method for estimating dimensionality and permits non-symmetric observation of task variables, a drawback of previous work. The introduction to the topic and summary of previous work was very clear and easy to read. Major Comments: In simulations the probability of neurons appearing on each trial was fixed at 40%. While the trial number was small relative to the number of neurons, lending support to the results, a value of 40% is not realistic for many neuroscience experiments. For example, for an experiment that includes dozens of sessions, the actual number may be closer to 1-5%. It would be useful to estimate how robust the analysis is under these conditions. Why was the non-sequential version of dPCA used? Is this decision more conservation than the sequential version because it allows dPCA to estimate the noise covariance? It’s hard for me to judge the difference in subspace angle error between MBTDR and dPCA. It seems that most of the additional error accrued by dPCA is in dimensions that already have a high error. Can you provide intuition why small errors in accurate dimensions supports your results? The ability to include structured prior covariances and noise covariance will likely yield interesting follow-ups with this method! -------- Author Response: I have read and am satisfied with the author's response.

Reviewer 3



The paper introduces an extension of an existing model, targeted dimensionality reduction (TDR), developed by Mante and colleagues in 2013. TDR is a regression based method to perform dimensionality reduction of neural data sets. Unlike PCA, which finds a basis with components sorted by the amount of total variance captured in the data, TDR identifies dimensions that specifically capture task-related variance, regardless of how much of the total variability they explain. This method assigns a single dimension per task variable, which as the authors explain, presents a clear limitation. There is no reason to believe that a given external covariate would elicit only one mode of activation in the neural population. Furthermore, this restriction implies that an important amount of variance in the data is potentially being missed. Therefore, by considering multi-dimensional task-related subspaces, and by identifying a low-rank factorization of the matrices defining such subspaces, the author's method MBTDR provides a means to discover additional structure in the population, which is of key importance in neuroscience. The features that MBTDR incorporates and are worth being highlighted are 1. They model is a probabilistic extension of TDR that defines a generative model of the data. This allows to incorporate a wide variety of assumptions, such as specifying different noise models or pre-defining structure using different prior settings. 2. The model can be fit to single trial data, even when there are unequal amounts of trials or missing observations. This is often the case in many data sets, as neurons are typically not simultaneously recorded, so the models are fit to the PSTHs, which are averaged responses. This averaging can destroy valuable information about the correlational structure in the population. 3. The authors provide a means of estimating the rank of the task subspaces. This is not straightforward as it involves searching over many possible rank configurations. The greedy algorithm proposed seems to be a reasonable approach to obtain optimal estimates for dimensionality. Similar methods like dPCA do not provide such a feature, although this can be accounted for post-hoc by keeping the number of dimensions needed to explain a certain percentage of the variance. This procedure, however, does not assess optimality. As the authors acknowledge, dPCA offers similar modelling advantages over TDR as MBTDR does. In particular dPCA is also able to identify multi-dimensional task subspaces. The authors go on by stating the limitations of dPCA with respect to their own method and compare the performance of the two models. They do so by testing how well the models recover the different parameter subspaces on generated data. However, given that the data was generated from the model itself -and in spite of the noise corruption-, I am not sure whether it is indeed surprising that their model can recover the subspaces better than dPCA. Similar arguments apply to figure 2A (although I guess this is partially addressed with the estimation under Poisson noise). Therefore, from the simulation studies, it was not clear to me whether MBTDR provides an advantage over dPCA at the task both dPCA and TDR were designed for: that of demixing variance from different task parameters. An example showing this would have been more illustrative, or at least a clearer reference to this in the text. I also could not really interpret this couple of sentences: "It is notable that the principle angle is an extremely sensitive measure of errors between subspaces and that both methods provide reasonable results when checked by eye. It is also notable that any differences are observable at all, which give us confidence that these results are quite strong." Finally, it is perhaps worth mentioning one limitation of these type of modelling approaches, which is that, no matter how much we increase the dimensionality, we are still dealing with a linear method, so non-linear relationships with respect to the external covariates cannot be captured. Regarding: -quality The model setup and fitting procedure seems rigorous. The authors used standard methods both for parameter estimation and rank estimation. However, I was not entirely familiar with some of the procedures and I did not check the math in detail, so I would not say I am confident about the methods in general. In any case, the authors seem to provide enough information to reproduce the study (provided that the code is released). -clarity The paper is well written, clear and easy to read. The figures are also good. I think most readers will appreciate figure 1A, nice visual illustration of the decomposition! -originality and significance The model is an extension of an existing method, so the study is not meant to be particularly original... but all the additional features that the model incorporates makes it an important contribution. minor clarification: line 219, SNR (in log10 units) is negative. These are high levels of noise, right? Any criteria for how this noise was set? how did parameter recovery depend on this? typos: line 169: TDA --> TDR line 192: decent --> descent line 300: a using (superfluous), dimension reduction —> dimensionality reduction figure 1B/2A: legend, EM --> ECME (if you wish...) In conclusion, I believe that the study is an important contribution to both neuroscience and machine learning. I did not identify any methodological flaws and the analysis seemed accurate. My only objection is whether the paper makes a strong enough neuroscientific point. First, regarding its superiority to identify neural structure and second, demonstrating that it is able to trade-off variance from different parameters -this last point if one wish to compare it to dPCA and TDR-. Except from this issue, which hopefully the authors can address, I would accept the submission with confidence. --------------------------------------------- Comments on authors' rebuttal The authors have satisfactorily addressed all the issues we raised. If the authors include additional analysis on one of the publicly available datasets from the dPCA eLife paper, that would resolve our biggest concern. Finally, making this method available to the neuroscience community will definitely lead to new insights when applied to different data sets and can guide the development of additional methods to analyse neural data.